# Selenium Yeast Alleviates Ochratoxin A-Induced Hepatotoxicity via Modulation of the PI3K/AKT and Nrf2/Keap1 Signaling Pathways in Chickens

**DOI:** 10.3390/toxins12030143

**Published:** 2020-02-25

**Authors:** Peng Li, Kang Li, Chao Zou, Cui Tong, Lin Sun, Zhongjun Cao, Shuhua Yang, Qiufeng Lyu

**Affiliations:** 1Key Laboratory of Zoonosis of Liaoning Province, College of Animal Science & Veterinary Medicine, Shenyang Agricultural University, Shenyang 110866, China; lipeng2018@syau.edu.cn (P.L.); 2017220529@stu.syau.edu.cn (K.L.); 13942686901@163.com (C.Z.); m18502463168@163.com (C.T.); sunny931230@163.com (L.S.); 2Tieling City Inspection and Testing and Certification Service Center (Animal Product Safety Testing Station), Tieling 112000, China; czj13470162988@163.com

**Keywords:** apoptosis, chickens, liver damage, ochratoxin (OTA), oxidative stress damages, selenium yeast

## Abstract

The aim of this study was to investigate the protective effects of selenium yeast (Se-Y) against hepatotoxicity induced by ochratoxin A (OTA). The OTA-induced liver injury model was established in chickens by daily oral gavage of 50 µg/kg OTA for 21 days. Serum biochemistry analysis, antioxidant analysis, as well as the qRT-PCR and Western blot (WB) analyses were then used to evaluate oxidative damage and apoptosis in chicken liver tissue. The results showed that Se-Y significantly increased liver coefficient induced by OTA (*P* < 0.05). OTA + Se-Y treated group revealed that Se-Y reduced the OTA-induced increase in glutamic pyruvic transaminase (ALT), glutamic oxaloacetic transaminase (AST) and malonaldehyde (MDA) content, and reversed the decrease in antioxidant capacity (T-AOC), glutathione peroxidase (GSH-Px) and total superoxide dismutase (T-SOD) (*P* < 0.05). In this study, we found that OTA is involved in the mRNA expression levels about Nrf2/Keap1 and PI3K/AKT signaling pathways, such as oxidative stress-related genes (Nrf2, GSH-Px, GLRX2 and Keap1) and apoptosis-related genes (Bax, Caspase3, P53, AKT, PI3K and Bcl-2). Besides, significant downregulations of protein expression of HO-1, MnSOD, Nrf2 and Bcl-2, as well as a significant upregulation of Caspase3 and Bax levels were observed after contaminated with OTA (*P* < 0.05). Notably, OTA-induced apoptosis and oxidative damage in the liver of chickens were reverted back to normal level in the OTA + Se-Y group. Our findings indicate that pretreatment with Se-Y effectively ameliorates OTA-induced hepatotoxicity.

## 1. Introduction

Ochratoxin A (OTA) is a toxic metabolite produced mainly by *Aspergillus* and *Penicillium* species of fungi and causes feed contamination to varying degrees at different stages. OTA has a negative impact on the health of livestock and causes huge economic losses [1]; therefore, greater control over its increased incidence in the food chain is urgently required [2]. Once ingested, OTA decomposition and the rate of its elimination is very slow, resulting in accumulation in tissues and organs [3]. OTA is targeted to the kidney [4], but also has hepatotoxic [5], neurotoxic [6,7], teratogenic and genotoxic effects [8], as well as reproductive toxicity [9]. As the main metabolic and detoxifying organ, the liver is an important target for most foreign compounds.

Studies have shown that even exposure to low concentrations of OTA in animals can lead to pathological and functional changes in the liver [10]. Thus, this toxin poses a huge challenge to human and animal health. The addition of anti-OTA dietary supplements has been proposed as a strategy to reduce oxidative stress (OS) damage. Oxidative stress occurs when the body is exposed to noxious stimuli that consequently produce a large number of reactive oxygen free radicals. The balance between the oxidation system and the antioxidant system becomes unstable when the oxide production capacity appears stronger than its scavenging capacity to lead to tissue damage [11,12]. Many antioxidants have been demonstrated to eliminate free radicals, providing cytoprotection against the OTA toxicity. Reports have demonstrated that glucurolactone (GA), silymarin (Sil), L-arginine (L-Arg), and compound ammonium glycyrrhizin (CAG) have hepatoprotective, anti-oxidative and anti-apoptotic effects in OTA-treated chicken primary hepatocytes [10]. Other antioxidants, such as quercetin, also have high anti-oxidant capacity, protecting Vero cells from OTA-induced damage [13]; catechins prevent OTA-induced LLC-PK1 cytotoxicity [14]; and rosmarinic acid has a protective effect on OTA-induced cytotoxicity in HepG2 cells [15]. As an essential trace element with strong anti-oxidant and anti-inflammatory effects, Selenium (Se) directly or indirectly affects the functions of various organs [16,17,18]. Se is known as “anti-hepatic necrosis protective factor” [19]. Se-Y is an organic form of Se that is more easily absorbed by animals. Since ingestion of Se-Y dietary supplements affects the regulation of oxidative defense, Se-Y is produced in industrial amounts compared to other antioxidants [17,20,21]. The identification of potential dietary supplements with better potency in their effects against OTA toxicity is challenging. However, the mechanisms underlying the detoxification and protective effects of Se-Y against OTA have not yet been reported in vivo.

The present study hypothesized that Se-Y protects against OTA-induced hepatotoxicity in chickens and that these effects are mediated by regulating the PI3K/AKT and Nrf2/Keap1 signaling pathways in the liver of chickens. More importantly, this is the first study to investigate the role of the PI3K/AKT and Nrf2/Keap1 signal paths in the protective effects of Se-Y against OTA-induced apoptosis and oxidative damage in the liver of chickens.

## 2. Results

### 2.1. Effects of Se-Y on OTA-Induced Changes in the Liver Index

Exposure to OTA resulted in a significant decrease in the liver index than that in the control group (*P* < 0.05; Figure 1). However, the liver index in the OTA + Se-Y group was significantly higher than that in the OTA group (*P* < 0.05). There were no significant differences in the liver indexes in the control, Se-Y and OTA + Se-Y groups.

### 2.2. Histopathological Changes in the Liver

HE staining showed that the hepatic lobule structure in the liver tissue of chickens in the control and Se-Y groups had clear intercellular boundaries and no inflammation, congestion, bleeding or exudation (Figure 2A,C). In the OTA group, some cells showed strong nuclear staining and marked inflammatory cell infiltration was observed around the blood vessels (Figure 2B). Furthermore, minor pathomorphological changes were revealed the liver sections from the OTA + Se-Y groups (Figure 2D). These results indicated that Se-Y administered as a dietary supplement exerted a protective effect against OTA-induced liver injury in chickens.

### 2.3. Analysis of Apoptosis by TUNEL

As shown in Figure 3A, the green fluorescence was particularly high in the OTA groups, which indicates that there was a large number of apoptotic cells in the livers of chickens. As shown in Figure 3B, the percentage of TUNEL positive cells was increased (*P* < 0.05) in the OTA group more than in the control group. More importantly, there was little the green fluorescence in the control, Se-Y and OTA + Se-Y groups, indicating that Se-Y had a certain inhibitory effect on apoptosis induced by OTA.

### 2.4. Effects of Se-Y on OTA-Induced Serum Biochemical Parameters

As biochemical indicators of liver damage, the levels of ALT and AST were determined to investigate the influences of OTA and Se-Y on liver injury. Compared with the control group, the activities of ALT and AST in the OTA group were significantly increased, while the levels were significantly decreased in the OTA + Se-Y group, compared to those in the OTA group (Figure 4; *P* < 0.05). These results indicated that administration of 50 µg/kg OTA daily for 21 days was associated with significant hepatotoxicity and that Se-Y protects against the toxic effects of OTA in the liver.

### 2.5. Effect of Se-Y on OTA-Induced Changes in the Oxidative Parameters

We analyzed the oxidative parameters of liver damage in each group. A significant increase in the liver MDA content, compared with the control group when exposed to OTA (Figure 5; *P* < 0.05). Furthermore, decreased levels of MDA were observed in the OTA + Se-Y, compared to the OTA group (*P* < 0.05). Activity of the anti-oxidant enzymes GSH-Px, SOD and T-AOC was decreased in the OTA group, compared with that in the control group (Figure 5; *P* < 0.05), while the levels of these enzymes were increased in the OTA + Se-Y group, compared with those in the OTA group (*P* < 0.05). These results suggest that Se-Y administered as a dietary supplement restored OTA + Se-Y to a level similar to that of the control group.

### 2.6. Nrf2-Keap1 Signaling and PI3K/AKT Signaling Pathway Gene Expression

To assess role of the PI3K/AKT and Nrf2/Keap1 signaling pathways in the protective effects of Se-Y against OTA-induced apoptosis and oxidative stress in the liver of chickens, we analyzed the relative mRNA expression levels of Keap1, Nrf2, MnSOD, GLRX2, GSH-Px, Caspase3, Bax, P53, AKT, PI3K and Bcl-2. In the OTA group, Keap1, Bax, Caspase3 and P53 were upregulated, while Nrf2, MnSOD, GLRX2, GSH-Px, AKT and PI3K mRNA were downregulated in the OTA group, compared to the levels detected in the control group (*P* < 0.05; Figure 6); however, there was no significant effect on the Bcl-2 expression between the two groups, although a downward trend was observed in the OTA group. In the OTA + Se-Y group, mRNA levels of Keap1, Bax, Caspase3 and P53 were downregulated, compared to the levels in the OTA group, while the levels of Nrf2, MnSOD, GLRX2, GSH-Px, AKT and PI3K were upregulated (*P* < 0.05; Figure 6). In addition, there were no significant differences in the control, the Se-Y, and the OTA + Se-Y groups. These results demonstrated that Se-Y has a potential protective effect on OTA-induced hepatotoxicity.

### 2.7. Effect of Se-Y on OTA-Induced Changes in Protein Expression

To determine whether exposure to OTA induced hepatotoxicity in the chickens, the protein levels of HO-1, MnSOD, Nrf2, Caspase3, Bax and Bcl-2 were measured by WB analysis (Figure 7). It was demonstrated that OTA treatment decreased expression of HO-1, MnSOD, Nrf2 and Bcl-2 proteins, while protein levels of Bax and Caspase3 were increased (*P* < 0.05). Compared to the OTA group, Nrf2, HO-1, MnSOD and Bcl-2 proteins were upregulated in the OTA + Se-Y group, while Caspase3 and Bax proteins were downregulated (*P* < 0.05). No significant differences were found in the protein levels of Nrf2, HO-1, MnSOD, Caspase3, Bax and Bcl-2 between the Se-Y and control groups. Taken together, our results provide further evidence that Se-Y could prevent OTA-induced toxicity in the liver of chickens.

The ratio of Bcl-2 and Bax has been proposed as an important factor in the regulation of apoptosis, with the ratio of Bcl-2 and Bax indicating increased apoptosis. Compared to the control group, a lower Bcl-2/Bax ratio was observed in the OTA group (Figure 7G). However, regarding the ratio of Bcl-2 and Bax, there were no significant differences between the control and OTA + Se-Y groups.

In conclusion, Se-Y exerted a protective effect against OTA-induced apoptosis and oxidative stress in the liver of chickens.

## 3. Discussion

In recent years, feed shortages have become a serious problem in livestock farming, and the resources available are often contaminated by OTA [22]. Therefore, the adverse effects of OTA-contaminated feed on livestock and the urgent need to find effective anti-OTA dietary supplements have received increasing attention. Golli Bennour et al. [23] have shown that OTA has a dose-dependent inhibitory effect on the viability of HepG2 human liver cancer cells. Studies also demonstrated that exposure of human hepatocarcinoma cells to OTA led to the induction of caspase-dependent apoptosis via the mitochondrial pathway. In the present study, we investigated the effects of Se-Y the liver of chickens fed with OTA-contaminated feed as an in vivo model. During the study, the clinical manifestations of the chickens were observed daily, and the chickens in the OTA group showed signs of depression, loose feathers, and white stools. After 21 days of treatment, the results showed that OTA reduced liver index. As shown in Figure 2, histopathological analysis revealed deep staining of some cells, and marked inflammatory cell infiltration around the blood vessels in the OTA group. Furthermore, the activity of ALT and AST was significantly increased in the OTA-treated group compared to that in the control group (Figure 3). This is consistent with the report by Mujahid et al. [24] showing that oral OTA also changed the different serum biochemical indicators of chickens. These results indicated that OTA causes liver damage in chickens, while Se-Y has protective effects on OTA-induced hepatotoxicity. Thus, this model provides a robust basis for further studies.

Oxidative stress refers to the imbalance of intracellular oxides and anti-oxidant systems, resulting in an oxidation-reduction equilibrium that tends to be in a peroxidized state, producing a large amount of ROS under conditions of internal and external stress [25,26]. Ineffective ROS removal results in damage to cellular lipids, proteins and DNA, which in turn impedes the cell signal transduction pathway [27,28]. T-AOC, GSH-Px and T-SOD are a major part of the endogenous anti-oxidant defense system and play a role in maintaining the intracellular redox balance [29,30,31]. MDA levels are a marker of cellular lipid peroxidation, which indirectly reflects the degree of cell damage [32]. In addition, Ansar et al. [33] showed that Se has a protective effect against oxidase activity. In accordance with this, we observed that the activity of GSH-Px, T-SOD and T-AOC in the liver of chickens exposed OTA were lower than those in the control group, while MDA levels were higher. Thus, our results further confirm that OTA-induced oxidative stress leads to liver damage. This study also demonstrated that the combination of Se-Y and OTA reduce MDA levels and restore the activity of GSH-Px, T-SOD and T-AOC, thereby enhancing the anti-oxidation systems in the liver. Thus, our results provide evidence that oral Se-Y improves tolerance to OTA-induced hepatotoxicity and maintains the redox balance in the liver of chickens.

Nrf2, which is a major transcription factor in the antioxidant system and is involved in the regulation of antioxidant enzyme expression, is inhibited by Keap1 under physiological conditions [34,35]. Previous studies have shown that OTA toxicity is associated with oxidative stress [36], apoptosis [37], autophagic cell death [38], mitochondrial dysfunction and induction of mitochondrial biogenesis [39]. Studies have indicated that oxidative stress is a key determinant of OTA-induced toxicity [40,41]. Nrf2/Keap1 is considered to be the most important therapeutic target for reducing oxidative damage caused by toxins [42]. We found that OTA exposure significantly affected the expression of oxidative stress-related genes, with elevated Keap1, while expression of Nrf2 and its target genes (MnSOD, GLRX2 and GSH-Px) showed a downward trend. It is clear that OTA leads to an imbalance in the antioxidant defense system of liver tissue. The combined treatment of Se-Y and OTA can increase the expression of liver antioxidant enzyme genes (GSH-Px, GLRX2 and MnSOD) and reduce liver damage; we hypothesized that OTA mimics the function of Keap1, preventing Nrf2 release from the Keap1 complex, or reduces Nrf2 protein expression in the cytoplasm by binding to Nrf2; however, the specific mechanism remains to be clarified. Protein levels of HO-1, MnSOD and Nrf2, in the livers of chickens contaminated with OTA were significantly lower than those in the control group (*P* < 0.05), indicating the existence of oxidative stress. Se-Y protected against the OTA-induced reduction in the levels of Nrf2, HO-1 and MnSOD proteins, indicating that Se-Y protects against OTA-induced oxidative stress (*P* < 0.05).

The PI3K/AKT pathway is widely present in various tissues and cells, and regulates the apoptosis, which is a form of programmed cell death, by inhibiting the activity of the Caspase family enzymes [9,43]. OTA induces apoptosis in neuronal cells [44] and the effects are likely to be restricted within particular structures of the brain [45]. Nrf2 is a downstream signaling protein regulated by the PI3K/AKT pathway and is inhibited by treatment with PI3K/AKT pathway-specific inhibitors [46,47]. Ramyaa et al. [13] showed that Nrf2-silencing resulted in increased expression of Caspase3 and Caspase9. The results of this study revealed that the expression of target genes and proteins associated with the PI3K/ATK signaling pathway was detected by qRT-PCR and WB, respectively. Compared with those in the control, increased expression of Caspase3, Bax and P53 was detected in the OTA group, while Bcl-2, AKT and PI3K expression was decreased. The Bcl-2 protein levels in the liver of OTA-treated chickens was significantly lower than those in the control (*P* < 0.05), while the levels of Caspase3 and Bax proteins were significantly higher, indicating the occurrence of apoptosis. In the OTA + Se-Y, we observed that Se-Y protected against the decrease in Bcl-2 protein levels and the increase in Caspase3 and Bax protein expression induced by OTA exposure.

## 4. Conclusions

In summary, our results show that Se-Y, with its high antioxidant potential, protects the liver of chickens from OTA-induced hepatotoxicity. More importantly, our findings may also have further implications for the amelioration of the toxic effects of other mycotoxins.

## 5. Material and methods

### 5.1. Reagents

OTA (purity > 98%) was provided by Pribolab (Immunos, Singapore); Se-Y (contained Se 2000 mg/kg) was obtained from Angel Yeast (Beijing, China) and was dissolved in autoclaved saline eluent. Kits for analyzing ALT, AST, MDA, GSH-Px, SOD and T-AOC contents were provided by Nanjing Jiancheng Bioengineering Institute (Nanjing, China). Total tissue RNA extraction kit, cDNA synthesis kit and qRT-PCR kit were acquired from Vazyme (Nanjing, China). Anti-Caspase3 (1:900, Abcam, Tokyo, Japan), anti-Bax (1:900, Immunoway, Suzhou, China), anti-HO-1, anti-Bcl-2, anti-Nrf2 (1:800, Bioss, Beijing, China), anti-MnSOD (1:6000, Enzo, Farmingdale, NY, USA), anti-actin (1:10,000, Abcam, Tokyo, Japan) and the horseradish peroxidase (HRP)-labeled goat anti-rabbit IgG (1:12,000, Jackson Immuno, PA, USA).

### 5.2. Animals and Treatments

Eighty healthy broiler chickens (aged 1 day) were purchased from a commercial rearing farm (Shenyang poultry farm, Liaoning province, China). The chicken cages, ground, and drink and feed containers in the facility were thoroughly cleaned and disinfected, and all utensils are fumigated. The chickens were then introduced to the facility and acclimatized for three days after transportation. The chickens were randomly allocated to the following groups (n = 20/group): control, 50 μg/kg OTA, 0.4 mg/kg Se-Y, and 50 μg/kg OTA + 0.4mg/kg Se-Y. The treatment was continued for 21 days. The doses of OTA and Se-Y were selected according to previous reports [48,49]. Chickens received feed and drinking water ad libitum for 21 days. The diets for each group were prepared simultaneously and stored in resealable bags prior to use. The following immunizations were performed according to routine procedures as follows: Day 7, the infectious bronchitis vaccine was inoculated by intranasal drip; day 14, the Newcastle disease vaccine was inoculated by drinking water, and the bursal vaccine was administered in drinking water. The chicken feed pellets complied with food hygiene standards. The animal experimental protocol was approved by the Ethics Committee, Shenyang Agricultural University, China (Permit No. 264SYXK < Liao > 2011-0001, date of approval: September 2018).

### 5.3. Collection of Samples

At 21 days of age, the liver was recorded. Blood samples were collected in tubes by puncture of the wing vein to obtain serum for AST and ALT testing. After taking blood samples, chickens were euthanized by cervical dislocation, and the liver was collected. Excess fat and the gallbladder were carefully removed from the liver, which was then blotted with a filter paper to remove the surface of the blood before the accurate weight was recorded. Subsequently, the liver was divided into two sections, one of which was fixed in 4% paraformaldehyde for hematoxylin and eosin (HE) staining, while the other section was placed in a numbered cryotube and stored at −79 °C prior to further use.

### 5.4. Histopathological Evaluation

The cut liver tissue was washed with saline tissue and immediately put into 4% formalin fixative solution (30–50 min), followed by 50% alcohol, 70% alcohol, 80% alcohol, 100% alcohol (0.5 h each time). Then transparent (to cover the lid to avoid moisture in the air), dipping wax (constant temperature), embedding, slicing (4–6 μm), adhesive sheet, dewaxing, dyeing, dewatering transparent and sealing (Seville Biotechnology, Wuhan, China).

### 5.5. TUNEL Apoptosis Analysis

The samples were prepared according to the instructions of the respective kit. Using paraffin-embedded kidney sections for TUNEL analysis (TUNEL was produced and analyzed by the China Seville Biotechnology Co., Ltd., Wuhan, China).

### 5.6. Detection of AST and ALT

The chicken blood was taken on an empty stomach, as soon as possible to separate the serum. Commercial kits AST and ALT were used in the experiment and according to the specification.

### 5.7. Analysis of the Oxidative Parameters of Liver

One g liver was accurately weighted and 9 times normal saline was added to make 10% homogenate. After centrifugation, the supernatant was carefully absorbed and diluted with normal saline to the optimum concentration. The supernatant of liver homogenate was used to measure MDA, SOD, T-AOC and GSH-Px.

### 5.8. Gene Expression Analyses

Total mRNA (the liver of chickens) was extracted using TRIzol (Vazyme, Nanjing, China). The total RNA purity and concentration were assessed according to the 260/280 nm absorbance ratio. The cDNA synthesis was performed with 1 µg total RNA using a PrimeScriptTMRT reagent kit (Vazyme, Nanjing, China).

Gene expression of β-actin, Nrf2, GSH-Px, GLRX2, Keap1, Bax, Caspase3, P53, AKT, PI3K and Bcl-2 was analyzed by qRT-PCR using the SYBR® Premix Ex TaqTM II kit (Vazyme, Nanjing, China) on an ABI Fluorescence Quantitative PCR Detection System (iQ5, ABI, Waltham, MA, USA). The analysis and primer pair synthesis (Table 1) were conducted by Sangon Biotech Co., Ltd. (Shanghai, China). All data were normalized to β-actin. Data were calculated by the 2^−ΔΔCt^ method.

### 5.9. Western Blot Analysis

Protein expression of HO-1, MnSOD, Nrf2, Capase3, Bax and Bcl-2 in the liver of chickens was determined by WB. Proteins were obtained from liver tissue using the total protein extraction kit (including RIPA and PMSF, Beyotime, Shanghai, China). The BCA protein assay kit (Solarbio, Beijing, China) was used to determine total protein content in the livers of the chickens. The proteins were resolved by SDS–PAGE gel and transferred onto the PVDF membranes (Solarbio, Beijing, China). Non-specific binding was minimized by incubation overnight at 4 °C with blocking buffer (5% skimmed milk powder). PVDF membranes were probed for 2 h at 4 °C with specific primary antibodies diluted in blocking buffer according to the dilution factors shown in Section 2.1. After washing six times with PBST, PVDF membranes were incubated for 60 min at 25 °C with HRP-conjugated secondary detection antibody diluted in blocking buffer according to the dilution factor shown in Section 2.1. Immunoreactive bands were visualized using the ECL (Beyotime Institute of Biotechnology, Shanghai, China) method. Band intensities were normalized against β-actin and relative intensities of bands were detected on a DNR Bio Imaging system (Ncmbio, Suzhou, China).

### 5.10. Statistical Analysis

All statistical tests were analyzed using the SPSS 19.0 software (IBM, Almon, NY, USA). Significant differences among the multiple groups were evaluated by one-way analysis of variance (ANOVA) with a post hoc test. *P* < 0.05 was considered a significant difference.

## Figures and Tables

**Figure 1 toxins-12-00143-f001:**
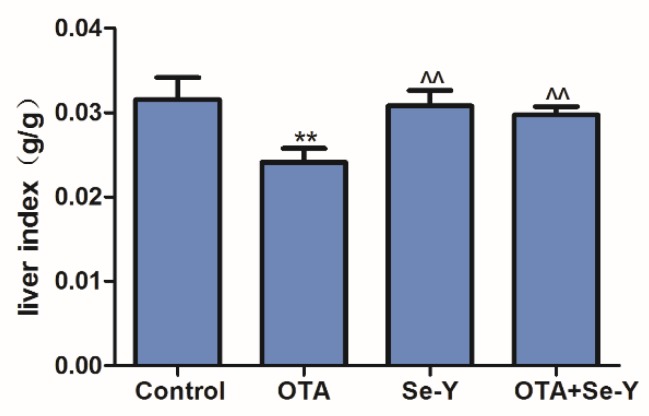
The liver index in the different groups. The organ coefficient = organ weight/body weight; Values represent mean ± SD (n = 6 chickens/group). ** *P* < 0.01 vs. control group, ^^ *P* < 0.01 vs. OTA group.

**Figure 2 toxins-12-00143-f002:**
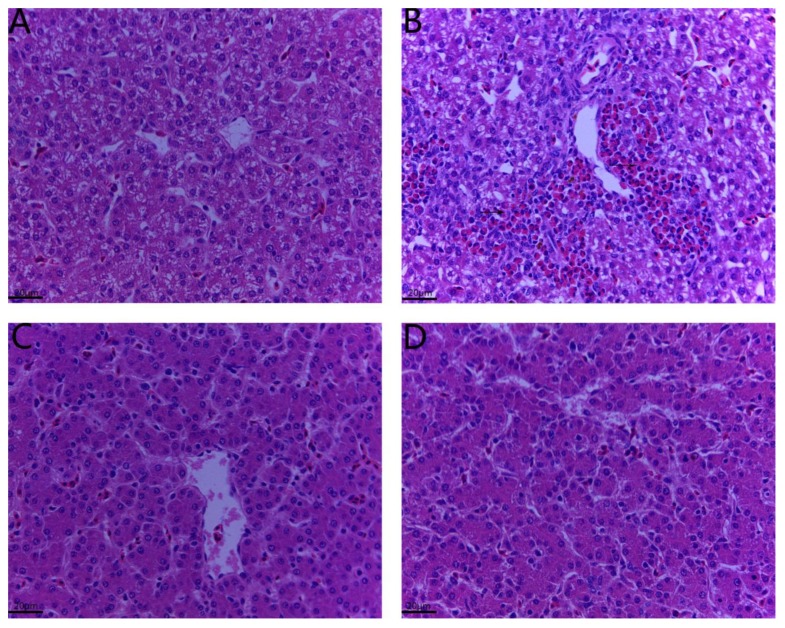
Detection of pathological changes in hepatic tissues by HE staining. (n = 6 chickens/group). (**A**) Control group, (**B**) OTA group, (**C**) Se-Y group, and (**D**) OTA + Se-Y group. The arrows “→” indicate a pathological injury in the livers (X400).

**Figure 3 toxins-12-00143-f003:**
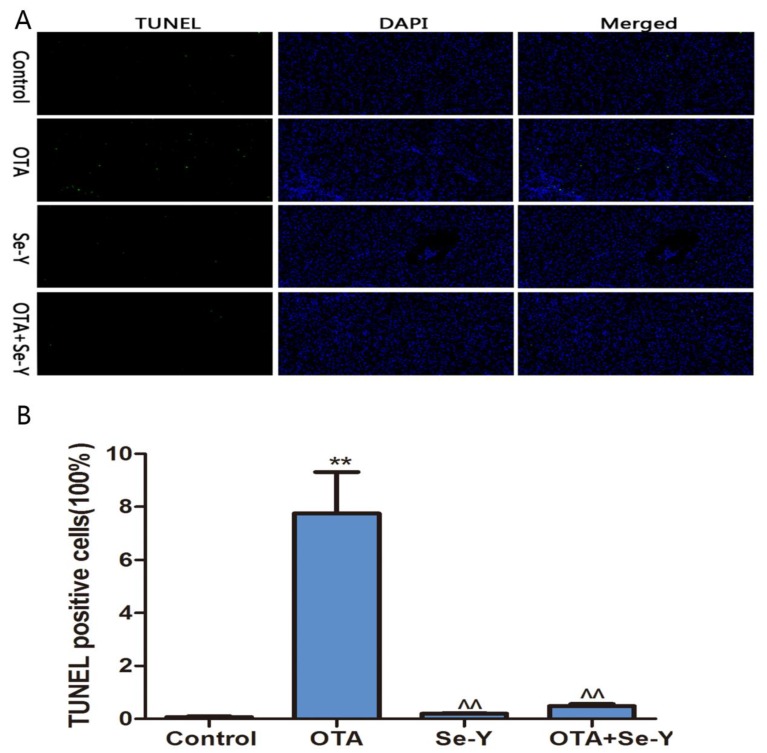
(**A**) TUNEL staining. Green fluorescence indicates TUNEL-positive cells. DAPI was used for nuclear staining (magnification 200×). (**B**) TUNEL positive cells (100%). (n = 6 chickens/group). ** *P* < 0.01 vs. control group, ^^ *P* < 0.01 vs. OTA group.

**Figure 4 toxins-12-00143-f004:**
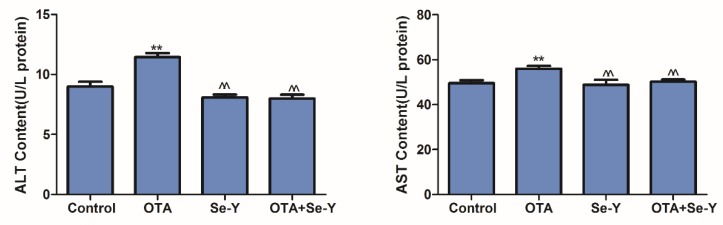
The activities of ALT and AST after the 21-day diet treatment in each treatment group. (n = 6 chickens/group). ** *P* < 0.01 vs. control group^^ *P* < 0.01 vs. OTA group.

**Figure 5 toxins-12-00143-f005:**
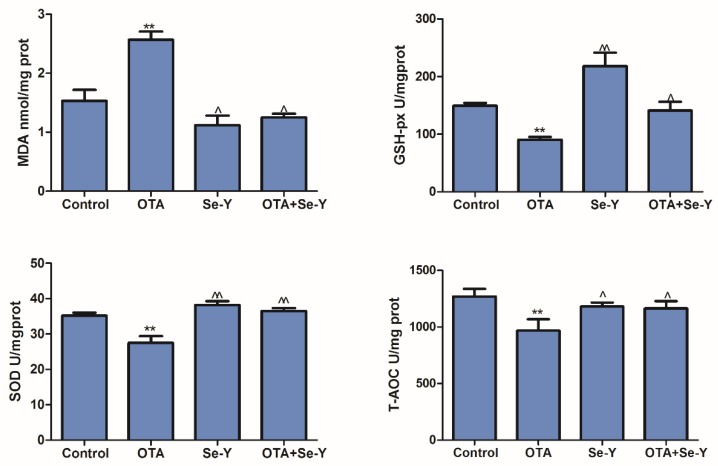
Oxidation and anti-oxidant parameters of hepatic tissues in each group detected using an oxidation kit. (n = 6 chickens/group). ** *P* < 0.01 vs. control group, ^ *P* < 0.05, ^^ *P* < 0.01 vs. OTA group.

**Figure 6 toxins-12-00143-f006:**
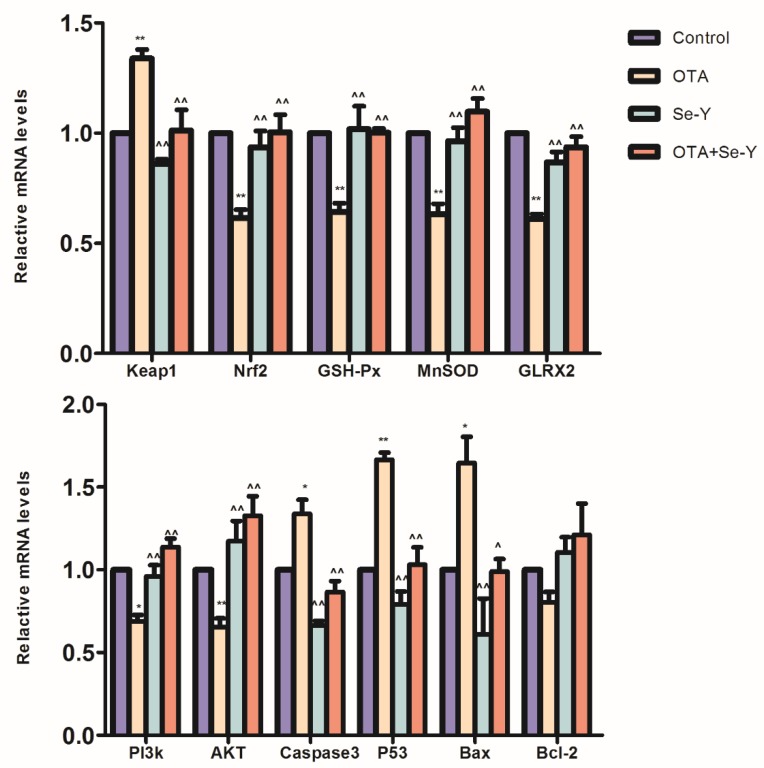
Nrf2/Keap1 and PI3K/AKT signaling pathways gene expression. (n = 3 chickens/group). * *P* < 0.05, ** *P* < 0.01 vs. control group, ^ *P* < 0.05, ^^ *P* < 0.01 vs. OTA group.

**Figure 7 toxins-12-00143-f007:**
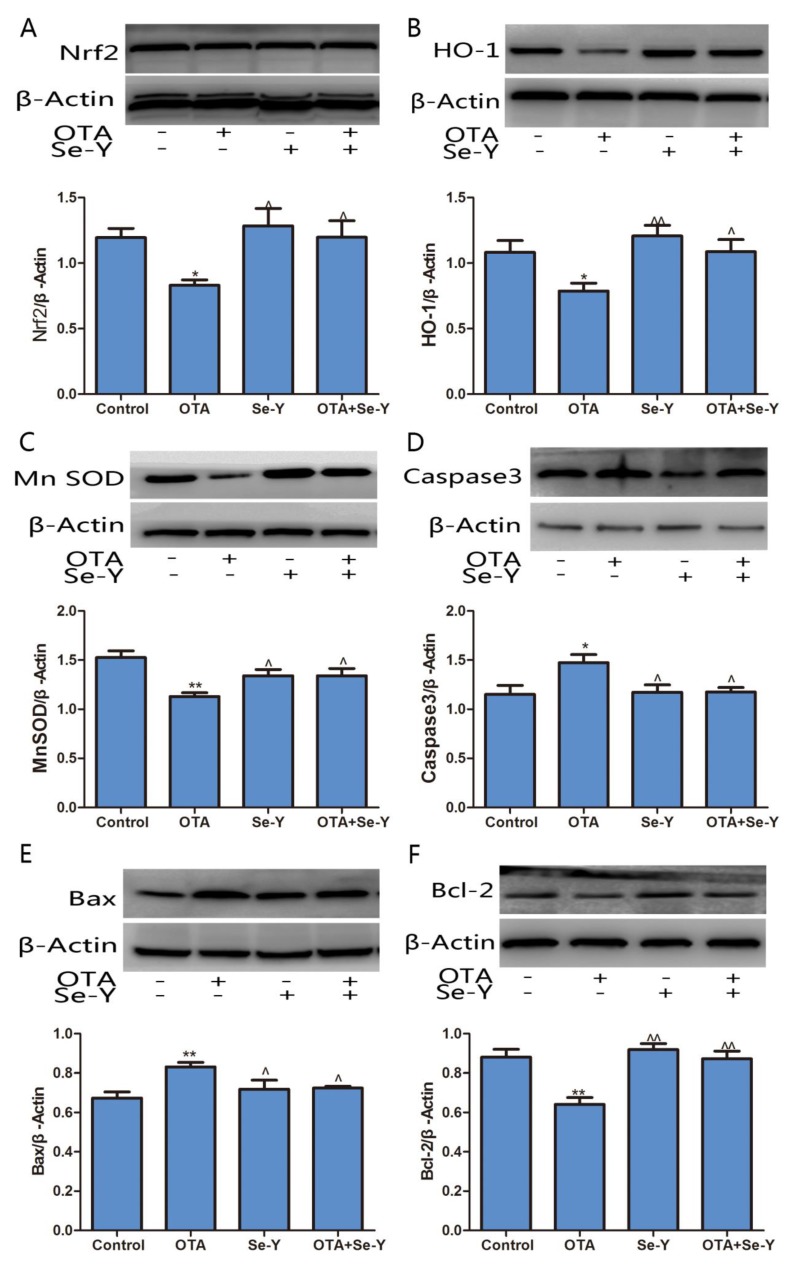
Effects of selenium yeast on the expression levels of protein involved in the Nrf2/Keap1 and PI3K/AKT signaling pathways that was induced by OTA in chicken livers. A:Nrf2, B:HO-1, C:Mn SOD, D:Caspase 3, E:Bax, F:Bcl-2, G: Ratio of Bcl-2 and Bax. (n = 3 chickens/group). * *P* < 0.05, ** *P* < 0.01 vs. control group, ^ *P* < 0.05, ^^ *P* < 0.01 vs. OTA group.

**Table 1 toxins-12-00143-t001:** The primer sequences of the target genes.

Primer	Sequence	Amplicon size (bp)
R-Nrf2-F: R-Nrf2-R: R-Keap1-F: R-Keap1-R: R-GSH-Px-F: R-GSH-Px-R: R-MnSOD-F: R-MnSOD-R: R-GLRX2-F: R-GLRX2-R: R-Bax-F: R-Bax-R: R-Bcl-2-F: R-Bcl-2-R: R-PI3K-F: R-PI3K-R: R-AKT-F: R-AKT-R: R-Caspase3-F: R-Caspase3-R R-P53-F: R-P53-R: R-β-actin-F: R-β-actin-R:	5’ CATAGAGCAAGTTTGGGAAGAG 3’ 5’ GTTTCAGGGCTCGTGATTGT 3’ 5’ ACTTCGCTGAGGTCTCCAAG 3’ 5’ CAGTCGTACTGCACCCAGTT 3’ 5’ CCAATTCGGGCACCAGGAGAA 3’ 5’ CTCTCTCAGGAAGGCGAACAG 3’ 5’ AAGGAGCAGGGACGTCTACA 3’ 5’ CCAGCAATGGAATGAGACCTGT 3’ 5’ ACGGAAGCCAGATCCAAGAC 3’ 5’ GTAGCACCTCCAACAAAAGACC 3’ 5’ GTGATGGCATGGGACATAGCTC 3’ 5’ TGGCGTAGACCTTGCGGATAA 3’ 5’ ATCGTCGCCTTCTTCGAGTT 3’ 5’ GTAGCACCTCCAACAAAAGA 3’ 5’ TACATTCTTGGGCTCCTT 3’ 5’ AGTGCGTGGAAATCTAAT 3’ 5’ AGTGCGTGGAAATCTAAT 3’ 5’ ATAATGACTATGGTCGTGC 3’ 5’ AAGCGAAGCAGTTTTGTTTGTG 3’ 5’ GCTAGACTTCTGCACTTGTCACCTC 3’ 5’ AAGCGAAGCAGTTTTGTTTGTG 3’ 5’ GCTAGACTTCTGCACTTGTCACCTC 3’ 5’ AGGAGAAGCTGTGCTACGTC 3’ 5’ TACCACAGGACTCCATACCCAA 3’	105 bp 142 bp 157 bp 82 bp 151 bp 91 bp 151 bp 170 bp 146 bp 128 bp 157 bp 183 bp

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
