# Peer review of "Selenium Yeast Alleviates Ochratoxin A-Induced Hepatotoxicity via Modulation of the PI3K/AKT and Nrf2/Keap1 Signaling Pathways in Chickens"

_toxins, 2020, doi:10.3390/toxins12030143_

Round 1

Reviewer 1 Report

Manuscript entitled „Selenium yeast alleviates ochratoxin A-induced hepatotoxicity via modulation of the PI3K/AKT and Nrf2/Keap1 signaling pathways in chickens” presents data from in vivo study which investigated the protective effect of selenium supplementation against toxic fungal metabolite. The presented data are in my opinion important and should be published. Generally, the manuscript is well written and I do not have any major comments, only some minor, as listed below:

Minor comments:

Introduction, line 31 – OTA is not produced by bacteria…please correct it. Also, please give the full name of OTA when mentioning for the first time. Figures 1,2,4,5 – please revise the titles (redundant repeat or some missing words, also title should be beginning with capital letter). Figure 3A – it is difficult to see any differences between pictures. Maybe increase the magnification if possible? Line 131 – “there were no significant difference in the control (…)” – I guess that other groups were compared to control? Results, all sections – please specify number of animals/observations for each type of analysis. Figure 7 instead of 7A. Line 179, 184, 188 - what is T-SOD? Earlier you mentioned SOD. Also, I did not find the explanations of the oxidative stress (give the full names when mentioning for the first time). Methods, line 293 – why b-actin was chosen as a reference gene? Did you check its expression was stable under experimental condition? Methods, line 301 – what was the amount of protein you used for Western Blotting? Statistics – which post hoc test was used? Language – please revise the whole manuscript (line 100: were was, etc.).

Author Response

Dear Reviewer:

Thank you for your letter and for the reviewers’ comments concerning our manuscript entitled. “Selenium yeast alleviates ochratoxin A-Induced hepatotoxicity via modulation of the PI3K/AKT and Nrf2/Keap1 signaling pathways in chickens”. (ID: 711372). Those comments are all valuable and very helpful for revising and improving our paper, as well as the important guiding significance to our researches. We have studied comments carefully and have made correction which we hope meet with approval. Revised portion are marked in red in the paper. The main corrections in the paper and the responds to the reviewer’s comments are as flowing:

Comment 1) Introduction, line 31 – OTA is not produced by bacteria…please correct it. Also, please give the full name of OTA when mentioning for the first time.

Response: Thank you for your review and suggestions, it has been modified in the article (Line 30-31).  “OTA is a toxic metabolite produced mainly by Aspergillus and Penicillium species of bacteria and causes feed contamination to varying degrees at different stages” were corrected as “Ochratoxin A (OTA) s a toxic metabolite produced mainly by Aspergillus and Penicillium species of fungi and causes feed contamination to varying degrees at different stages”.

Comment 2) Figures 1,2,4,5 – please revise the titles (redundant repeat or some missing words, also title should be beginning with capital letter).

Response: We are so sorry for the mistake in the article. It has been modified in the article: Figure 1 (Line 74), Figure 2 (Line 87-88), Figure 3 (Line 99), Figure 4 (Line 111) and Figure 5 (Line 124). I appreciate your carefulness very much.

Comment 3) Figure 3A – it is difficult to see any differences between pictures. Maybe increase the magnification if possible?

Response: Thank you for your review and suggestions, it has been modified in the article (Line 86).

Comment 4) Line 131 – “there were no significant difference in the control (…)”

Response: “There was no significant effect on the Bcl-2 expression between the two groups although a downward trend was observed in the OTA group.” In general, when the experimental result is less than the 0.05 level or the 0.01 level, it means that there is a significant difference between the control and the OTA groups. There was no statistically significant difference in Bax gene, but there was a downward trend, which also indicated that OTA caused apoptosis in chicken  liver.

Comment 5) Results, all sections – please specify number of animals/observations for each type of analysis. Figure 7 instead of 7A.

Response: Thank you for your review and suggestions, it has been modified in the article.

Comment 6) Line 179, 184, 188 - what is T-SOD? Earlier you mentioned SOD.

Response: In line 14, “superoxide dismutase (T-SOD)” was corrected as “total superoxide dismutase (T-SOD)”. T-SOD is a free radical scavenger, which can remove toxic superoxide anion free radicals.

Comment 7) Also, I did not find the explanations of the oxidative stress (give the full names when mentioning for the first time).

Response: Thank you for your review and suggestions, it has been modified in the article (Line 42-46). Oxidative stress (OS) occurs when the body is exposed to noxious stimuli that consequently produce a large number of reactive oxygen free radicals. The balance between the oxidation system and the antioxidant system becomes unstable when the oxide production capacity appears stronger than its scavenging capacity to lead to tissue damage (Line 42-46).

Comment 8) Methods, line 293 – why b-actin was chosen as a reference gene? Did you check its expression was stable under experimental condition?

Response: Actin is an important backbone protein for cells. β-Actin is a commonly used internal reference for qRT-PCR. Its expression in tissues and cells is relatively constant. It is often used as a reference when detecting changes in gene expression levels.

Comment 9) Methods, line 301 – what was the amount of protein you used for Western Blotting?

Response: Protein expression of HO-1, MnSOD, Nrf2, Capase3, Bax and Bcl-2 in the liver of chickens was determined by WB. Among them, the protein mass used in the determination of HO-1, MnSOD, Nrf2, Bax and Bcl-2 was 30ug. However, the expression of Capase3 was low in frozen tissues. By exploring the best protein mass, the final protein mass was determined to be 50ug.

Comment 10) Statistics – which post hoc test was used?

Response: Significant differences among the multiple groups were assessed by one-way analysis of variance (ANOVA).

Comment 11) Language – please revise the whole manuscript (line 100: were was, etc.).

Response:Thank you for your review and suggestions, it has been modified in the article. Our manuscript have been checked by a native English speaking colleague.

Reviewer 2 Report

Manuscript title: Selenium yeast alleviates ochratoxin A-Induced hepatotoxicity via modulation of the PI3K/AKT and Nrf2/Keap1 signaling pathways in chickens

General comments:
The aims of the submitted study were focused in the investigation of the protective effects of selenium yeast (Se-Y) against hepatotoxicity induced by ochratoxin A (OTA). In general, the authors have completed a reasonable study with very informative data on the animal study of OTA-induced damages. However, some of minor concerns would be suggested and requested for further improvement in the manuscript.

The following points may be considered while revising the article:

Specific Comments:

Please check the contents in Line 249-252. It seems not the exact experiments in the study. The proximate composition analysis of selenium yeast used in the study should be analyzed or described in the study, but not only by the description of the purity > 99.5%. The complicated compositions existed in Se-Y should be not a pure compound. The format of cited references was not in full compliance with the Journal format. Please check for accuracy and correct if wrong.

Author Response

Dear Reviewer:

Thank you for your letter and for the reviewers’ comments concerning our manuscript entitled. “Selenium yeast alleviates ochratoxin A-Induced hepatotoxicity via modulation of the PI3K/AKT and Nrf2/Keap1 signaling pathways in chickens”. (ID: 711372). Those comments are all valuable and very helpful for revising and improving our paper, as well as the important guiding significance to our researches. We have studied comments carefully and have made correction which we hope meet with approval. Revised portion are marked in red in the paper. The main corrections in the paper and the responds to the reviewer’s comments are as flowing:

 Comment 1) Please check the contents in Line 249-252. It seems not the exact experiments in the study.

Response: Thank you for your review and suggestions. This is a normal immunization program to prevent chicken infectious bronchitis vaccine, chicken Newcastle disease and chicken infectious bursal disease.

Comment 2) The proximate composition analysis of selenium yeast used in the study should be analyzed or described in the study, but not only by the description of the purity > 99.5%. The complicated compositions existed in Se-Y should be not a pure compound.

Response: Thank you for your review and suggestions, it has been modified in the article (Line 245-246).

Comment 3) The format of cited references was not in full compliance with the Journal format. Please check for accuracy and correct if wrong.

Response: Thank you for your review and suggestions, it has been modified in the references.

Reviewer 3 Report

This article should be accepted for the publication in Toxins. The study is very interesting and good quality. Noteworthy is a wide range of research, an innovative approach to the topic under study, very good presentation and interpretation of the results in both text and graphics. The obtained results provide a solid basis for deepening and continuing the research topic.

Author Response

Dear Reviewer:

Thank you for your letter and for the reviewers’ comments concerning our manuscript entitled. “Selenium yeast alleviates ochratoxin A-Induced hepatotoxicity via modulation of the PI3K/AKT and Nrf2/Keap1 signaling pathways in chickens”. (ID: 711372). On behalf of my co-authors, we thank you very much for giving us an opportunity to accept our manuscript.

Thank you and best regards.

Reviewer 4 Report

Nice experimental study of potential clinical relevance. I have a few minor points.

Minor points:

Include in the abstract few results with SEM or SD plus p valuues of the most important parametuer(s) of your study. Under 2.2, line 74: reword: was clear. with clear. In the discussion, amplify the situation in humans and quote papers of human liver injury by contaminants ingested.   Line 79. avoid the term dietary supplement, wrong word, check literature.

Author Response

Dear Reviewer:

Thank you for your letter and for the reviewers’ comments concerning our manuscript entitled. “Selenium yeast alleviates ochratoxin A-Induced hepatotoxicity via modulation of the PI3K/AKT and Nrf2/Keap1 signaling pathways in chickens”. (ID: 711372). Those comments are all valuable and very helpful for revising and improving our paper, as well as the important guiding significance to our researches. We have studied comments carefully and have made correction which we hope meet with approval. Revised portion are marked in red in the paper. The main corrections in the paper and the responds to the reviewer’s comments are as flowing:

Comment 1) Include in the abstract few results with SEM or SD plus p valuues of the most important parametuer(s) of your study.

Response: Thank you for your review and suggestions, it has been modified in the abstract.

Comment 2) Under 2.2, line 74: reword: was clear. with clear.

Response: Thank you for your review and suggestions, it has been modified in the article (Line 78). Line 78, the statements of “HE staining showed that the hepatic lobule structure in the liver tissue of chickens in the control and Se-Y groups was clear, with clear intercellular boundaries and no inflammation, congestion, bleeding or exudation (Fig. 2A,C).” were corrected as “HE staining showed that the hepatic lobule structure in the liver tissue of chickens in the control and Se-Y groups was clear intercellular boundaries and no inflammation, congestion, bleeding or exudation (Fig. 2A,C).”

Comment 3) In the discussion, amplify the situation in humans and quote papers of human liver injury by contaminants ingested.

Response: Thank you for your review and suggestions, it has been modified in the article. We have added "Golli Bennour et al [21] have shown that OTA has a dose-dependent inhibitory effect on the viability of HepG2 human liver cancer cells. Study also demonstrated that exposure of human hepatocarcinoma cells to OTA led to the induction of caspase dependent apoptosis via the mito¬chondrial pathway. " in the paper.

Comment 4) Line 79. avoid the term dietary supplement, wrong word, check literature.

Response: Thank you for your review and suggestions. The US Food and Drug Administration (FDA) has announced the Dietary Supplement Health and Education Act (DSHEA), which sets out the following requirements for dietary supplements. A product intended to supplement the diet, it may contain one or more of the following dietary ingredients: a vitamin, a mineral, an herb, an amino acid, a food ingredient that supplements a diet, or a concentrate, metabolite, ingredient, extract, or combination of the above.
